# Does Temporal and Spatial Diet Alteration Lead to Successful Adaptation of the Eastern Imperial Eagle, a Top Predator?

Dimitar Demerdzhiev [1,2,*], Zlatozar Boev [2], Dobromir Dobrev [1], Nedko Nedyalkov [2] and Tseno Petrov [1]

1   Bulgarian Society for the Protection of Birds, Nikola Vaptsarov Str. 3, Asenovgrad/BirdLife Bulgaria, Leonardo Da Vinci Str. 5, 4000 Plovdiv, Bulgaria
2   National Museum of Natural History, Bulgarian Academy of Sciences, Blvd. Tsar Osvoboditel 1, 1000 Sofia, Bulgaria
*   Correspondence: dimitar.demerdzhiev@bspb.org

**Abstract:** Predator–prey interactions may be linked to different temporal or spatial patterns, including dynamics in prey populations. Therefore, understanding the adaptive capacity and how top predators respond to shifts in prey abundance and availability is crucial for their conservation. In this study, we investigated the diet pattern of the endangered Eastern Imperial Eagle facing long-term and large-scale changes. We studied the abundance variation of its profitable prey, sousliks, and how it reflected on eagle population trajectories in a regional and temporal context. We found a significant diet alteration expressed in large decrease of brown hare ($\beta^2 = -0.83$), poultry ($\beta^2 = -0.81$), gulls ($\beta^2 = -0.71$), and water birds ($\beta^2 = -0.57$), and an obvious increase of northern white-breasted hedgehog ($\beta^2 = 0.61$) and doves ($\beta^2 = 0.60$). Raptors and owls raised their participation ($\beta^2 = 0.44$), but white stork and different reptiles supplied more biomass. Abundance of European souslik decreased through the studied periods (adjusted $R^2 = 0.25$, $p < 0.001$) which accounted for the lower proportion of this prey in the eagle's diet. Nevertheless, the eagle population successfully adapted and significantly increased ($\beta^2 = 0.97$) in most of the distribution area. The trophic strategy used by this top predator related to opportunistic foraging represents an ecological advantage that allows the species to adapt to different habitats and guarantees its future. The observed prolonged diet alteration could result in a significant negative attitude among different groups such as hunters, pigeon fanciers, and poultry keepers towards eagles. Therefore, enhanced communication with key stakeholders is needed. Conservation efforts should be focused also on the preservation of the species' main foraging habitats and the restoration of damaged ones so as to maintain the good conditions of both primary food source and subsequent prey.

**Keywords:** *Aquila heliaca*; food spectrum; diet changes; diurnal raptors; long-term studies; generalist; adaptation; prey

## 1. Introduction

Large raptor species are limited by different factors such as food supply, nest-site availability, weather conditions, and bird experience [1–3]. Generalist predators can change their diet mainly in response to habitat alteration and depletion of main food resources. Such changes in diet can affect population trajectories via individual fitness and breeding performance [4,5]. Therefore, understanding the adaptive capacity and how top predators respond to shifts in prey abundance and availability is crucial for their conservation. Successful adaptation of top predators to changes in availability and abundance of main prey sources determines their ability to survive and expand their populations in a changing environment.

It is generally considered that generalists are more adaptable to spatially or temporally heterogeneous environments, while specialists are more adaptable to temporally stable environments [6]. Since generalist species may have wider dietary niches and can switch

between prey resources, when the preferred prey declines spatially or temporally, they are less susceptible to the negative demographic effect caused by changes in prey availability and abundance than more specialized species [7], but see 39. However, the classification of generalist or specialist can occur along a gradient of adaptability and, furthermore, a generalist species can be made up of specialized individuals ([8], as well as a typical "specialist" can successfully adopt a generalist foraging strategy [9]. Then, within a species range, the individual's capacity to utilize alternative resources is crucial for successful adaptation when the main prey is depleted. In fact, the response at the individual level may vary depending on how individuals rank their prey, which, in turn, results in different resource use patterns [10–12].

Processes in ecology vary over time [13], and predator–prey interactions may be linked to different temporal or spatial patterns, including cycles and outbreaks in prey populations [14,15], leading to spatial and temporal shifts in the predator's diet [16,17]. Therefore, in this predator–prey system, the predator may include new alternative prey sources when preferred prey is scarce [18,19].

Here, we examine a generalist predator, the eastern imperial eagle (*Aquila heliaca*), hereafter EIE, foraging in open habitats with predominant grass vegetation [20], where it exploits various prey species of different size [21–26]. Different souslik species (*Spermophilus* sp.) represent profitable prey of this eagle, determining the distribution and density of the largest Eastern populations in Russia and Kazakhstan [26–28]. However, in other parts of the species' distribution area, the lack of large continuous souslik colonies leads to dietary shifts and wider prey diversity [21–25,29]. Previous studies recorded regional diet differences in terms of subpopulations [24,26], and those differences were strongly influenced by the individual territories occupied by the eagles [22]. Temporal changes in the EIE diet are well-documented only for the westernmost Pannonian population, where traditional prey species such as common hamster (*Cricetus cricetus*) and European souslik (*Spermophilus citellus*) are shifted by corvids (*Corvidae*), water birds, and roe deer (*Capreolus capreolus*) [24].

In this study, we investigated if the diet pattern of an endangered top predator such as the EIE faced long-term and large-scale changes, and if so, how the eagle responded to such shifts. We studied the abundance variation of its profitable prey, such as sousliks, and how it reflected on the EIE population trajectories in a regional and temporal context.

Our aim was to explore the adaptive capacity of this generalist species describing the mechanism of changes in the resource use pattern by which it switched between the different food sources.

We predicted that if the availability and abundance of profitable prey decreased, eagles could substitute the decreasing prey with other plentiful food sources and thus survive and increase their number, and vice versa—the lack of sufficiently abundant and accessible prey would lead to territory abandonment and population decline. We studied which species and to what extent could substitute the decreasing prey and planned future conservation strategies.

## 2. Materials and Methods

### 2.1. Study Area

The diet remains were collected in the whole distribution area of the species in Bulgaria [22]. We sampled 37 different breeding territories, distributed among six geographical units (Figure 1). Mountainous habitats (ER, SG) were characterized by small fragments of pastures and meadows and considerable forest cover, where eagles bred close to the forest edge, using common beech (*Fagus sylvatica*), sessile oak (*Quercus petraea*), and scots pine (*Pinus silvestris*) [30]. We merged territories from SG (n = 2) and ER (n = 1) into a group of high mountain regions (HM) due to the small sample sizes and the similar habitat conditions [22]. The EIEs in other regions occupied hilly areas and lowlands, where grasslands, usually overgrown with shrub formations of oriental hornbeam (*Carpinus orientalis*) and Christ's thorn (*Paliurus spina-christi*), agricultural fields and small forest patches formed

a mosaic habitat structure. They built their nests on single trees or in small groups of trees, mainly hybrid poplars (*Populus* sp.) or Hungarian oak (*Quercus frainetto*), downy oak (*Quercus pubescens*), and Turkey oak (*Quercus cerris*), often along small streams or in fields.

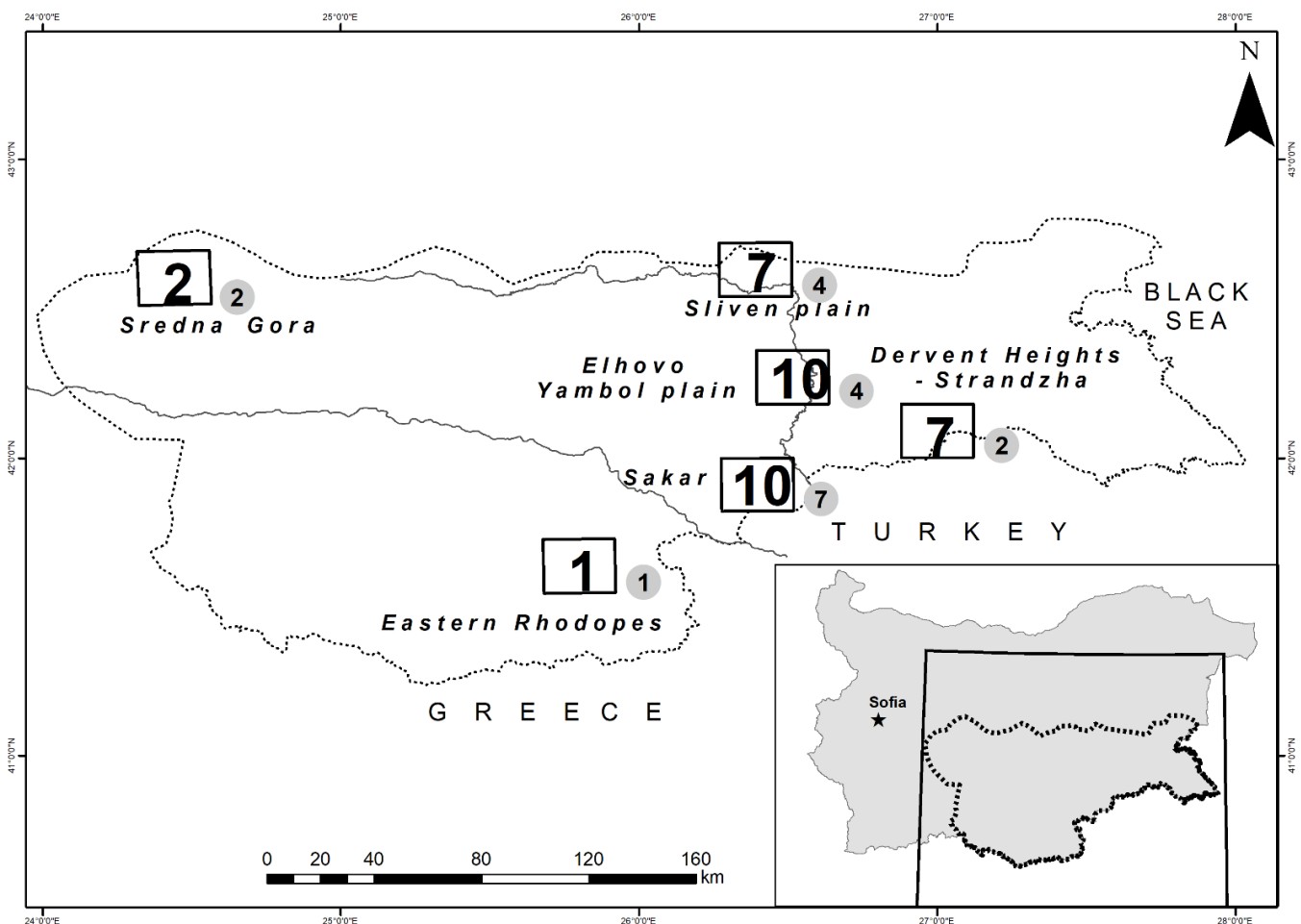

**Figure 1.** Number of sampled breeding territories of the eastern imperial eagle (in a square) vs. number of sampled souslik plots (in circle) in the different regions (Eastern Rhodope Mnt., ER; Sredna Gora Mnt., SG; Sliven plain, SP; Elhovo-Yambol plain, EYP; Dervent Heights-Western foothills of Strandzha Mnt., DHWstr; Sakar Mnt.).

### 2.2. Data Collection

In this study, we used a 23 years' data set (1999–2021), part of which has already been published, although in a different context [22]. The first detailed study on the diet of the EIE from Bulgaria considered only food composition, regional distribution, and seasonal differences, not reporting the temporal changes [22]. In this study, we analyzed the temporal variation in main prey species, using data about 5315 prey specimens, covering the entire distribution area of the EIE in the country (Table 1). The annual number of sampled EIE territories corresponded to the number of occupied territories ($r_c$ = 0.81, $p$ = 0.0001) (Figure 2).

Each nesting site was visited twice in each of the following periods: November–February, June–August (post-fledging period). Food remains, bones, feathers, and pellets were collected inside and under nests and roosts [31]. The following types of remains were not included in the data in order to reduce the bias of indirect sampling, even if they were found under the nest sites or roosting trees: (1) single feathers, which could be shed by live birds; (2) full carcasses of large animals, which could not be brought there by the eagles; (3) old or deteriorated samples, which could have remained from previous years [22,24]. The material was identified through the comparative osteological collections of the National

Museum of Natural History at the Bulgarian Academy of Sciences. Whenever possible, the minimum number of individuals (MNI) in each pellet or prey remain was estimated based on the number of skeletal or keratinized body parts [22,31]. The MNI was determined by taking into account the age (juvenis, subadultus, adultus), sex, and the size differences between individuals.

**Table 1.** Number of sampled prey specimens and number of sampled eastern imperial eagle occupied territories in different regions.

| Regions | Number of Sampled Occupied Territories | Number of Sampled Prey Specimen |
|---|---|---|
| HM (Sredna Gora Mnt. and Eastern Rhodope Mnt.) | 3 | 456 |
| Sakar Mnt. | 10 | 2090 |
| DHWstr (Dervent Heights-Western foothills of Strandzha Mnt.) | 7 | 1239 |
| EYP (Elhovo-Yambol plain) | 10 | 916 |
| SP (Sliven plain) | 7 | 614 |
| **TOTAL** | **37** | **5315** |

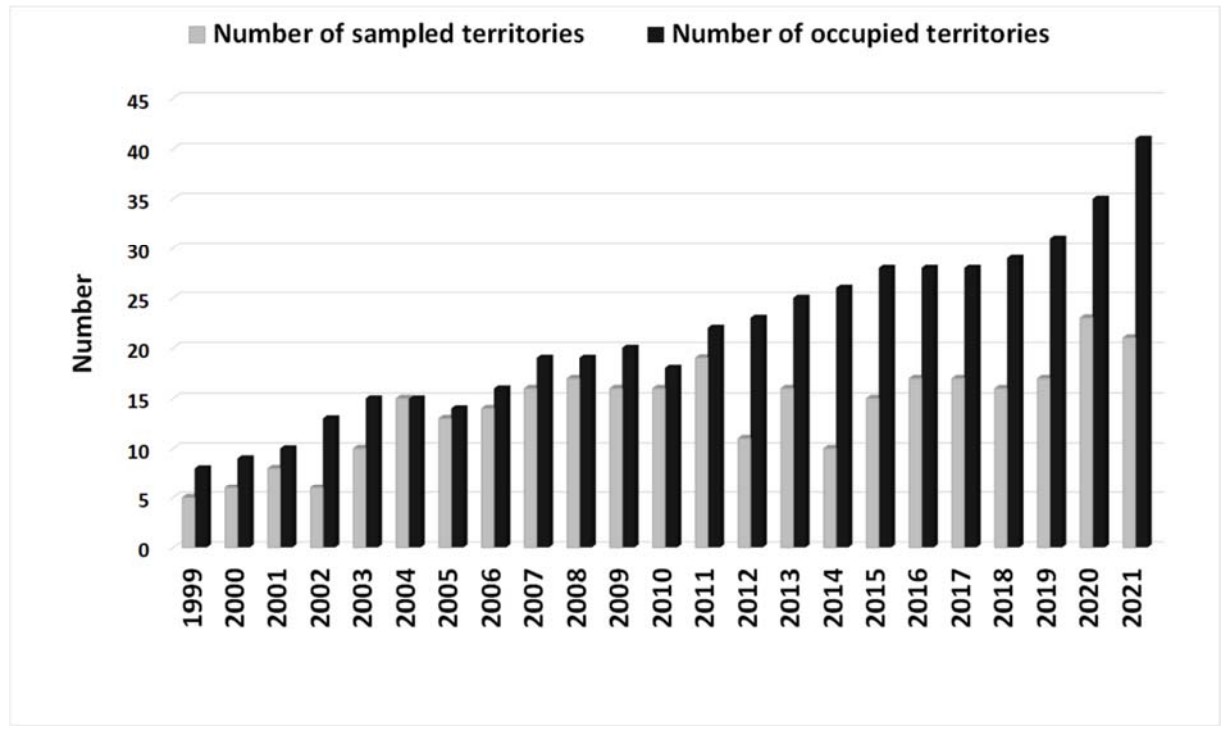

**Figure 2.** Number of sampled eastern imperial eagle territories and number of occupied territories in different years.

The body mass of the specimens of the various species was determined by [32–39]. An average body mass was given, calculated on the basis of the average mass of individual specimens. When the material was identified up to genus level, the average values for the presented species of the genus were given. The carrion biomass was not taken into account [22].

### 2.3. Profitable Prey Abundance

Prey abundance of profitable prey (European souslik) was estimated through test plots, each covering 1 ha, where all active holes were counted and recorded [40]. In total, 20 such plots were monitored during the whole study. The plots were located in the souslik colonies

distributed in all the studied regions within the EIE's occupied territories (Figure 1). To avoid the effect of cycles in rodent abundance between years, each plot was visited in two consecutive years of the three study periods (see below). In peak years, rodents would be highly abundant, while in the poor years, the opposite will be valid [3]. The count of the abundance of sousliks was carried out twice in the studied year, in the months of April and May. The mean value of the reported individuals (active holes) per plot in a given year was taken in the analysis.

*2.4. Data Analyses*

In order to identify the main changes in the diet composition, the prey items were grouped into the main categories, following the already published methodology [22]: lizards and snakes (Squamata), tortoises (Testudines), water birds (Anatidae, Ardeidae), poultry (*Gallus gallus f. domestica*, *Anser anser f. domestica*, *Meleagris gallopavo f. domestica*, *Pavo cristatus f. domestica*), phasianids (Phasianidae), gulls (Laridae), doves (Columbidae, Feral Pigeon), songbirds (Non-Corvidae Passerines), corvids (Corvidae), stork (*Ciconia ciconia*), raptors and owls (Accipitridae, Falconidae, Strigidae, Tytonidae), hedgehog (*Erinaceus roumanicus*), hare (*Lepus europaeus*), souslik (*Spermophilus citellus*), rodents (Rodentia excl. European souslik), carnivores (Carnivora), carrion (Artiodactyla, Perissodactyla), and other animals (*including other vertebrate taxa*).

We divided our data set into three periods associated with significant changes in the country's land use pattern that might have affected the populations of eagle's prey [41]. The first period (1999–2006) included the years prior to Bulgaria's accession to the European Union (EU). This period was characterized by extensive agriculture and animal husbandry. The second period (2007–2013), related to the country's accession to the EU, was characterized by gradual intensification of some aspects of agriculture through subsidies. During this period, intensive plowing of natural and semi-natural grasslands and their conversion back to arable land was registered [41]. The third period (2014–2021) included the last programming period of the European Commission's CAP (Common Agriculture Policy). It was characterized by a new habitat alteration financially stimulated by subsidies and expressed in large-scale removal of shrubs from grasslands using mechanized equipment such as shredders and bulldozers. We compared the frequency and biomass contribution of the different prey categories among the geographical units and in general during the three periods in order to investigate if there were any evident long-term and large-scale alterations in the diet composition.

As a first step, we applied the over-parameterized linear model (GLM) with Type III error distribution. We ran two models: one for the proportion and one for the biomass, including the year as a continuous covariate and each prey category as explanatory factors. Secondly, we built a simple mixed model (GLMM) including the prey category and the study period as explanatory factors. To control for spatial variation in prey abundance and composition, as well as for possible differences in feeding strategies among eagle populations living in different environments, we included a random factor "region" to account for data pooled within each region. We ran two models again: one for the proportion and one for the biomass. After that we applied a post-hoc analysis (Tukey's HSD test) to extract the significance of the trends of each category.

Changes in profitable prey abundance (souslik density) were evaluated through GLMM where souslik density and study periods were explanatory variables and "region" was included as a random effect. A *post hoc* analysis (Tukey's HSD test) was used to extract the significance of the trends of souslik density in each period from the model.

The design we used could not account for the fact that samples collected in the same nest or nearby trees could be predated in the different years by the same individuals [24]. Eagles breeding in a given nesting site could change over the years, then remains collected in the same nesting sites could derive from independent individuals. Similarly, the remains in a nest in a particular year included items predated by the male or the female of the given pair in an unknown proportion; therefore, the data about the two individuals could



not be separated in the individual samples. Hence, the factor "individual eagle" could not be included in our model. Nonetheless, we considered our aim to detect long-term and large-scale changes in the eagles' diet achieved because of the large applicable and representative data sampling.

The data calculated in percentage (prey frequency and biomass contribution) were converted into proportions and then Arcsin transformed to achieve a close to normal distribution [42]. To evaluate the results of the regression models, we used the adjusted $R^2$ value as a correction factor. We also used explanatory parameter estimates ($\beta^2$) with lower and upper CL (95%) and a probability value (*p*) of the explanatory factors. Results with $p \leq 0.05$ were considered significant. Values were provided as means $\pm$ standard error (SE). All data were analyzed using Statistica for Windows, Release 12 [43].

## 3. Results

### 3.1. General Pattern of Main Prey Contribution and EIE Population

The 23 years' trend of the different prey categories showed a significant EIE diet alteration (Table 2). While the share of hare, poultry, and gulls showed the largest decrease, both in terms of frequency and biomass, hedgehogs and doves increased significantly their presence and biomass contribution (Table 2, Figure 3). To a lesser extent, water birds also reduced their occurrence (adjusted $R^2 = 0.29$, $\beta^2 = -0.57$, $p = 0.004$) and biomass supply (adjusted $R^2 = 0.14$, $\beta^2 = -0.42$, $p = 0.047$). Decline was found also for the presence of carrion (adjusted $R^2 = 0.35$, $\beta^2 = -0.61$, $p = 0.002$) (Figure 3). In contrast, raptors and owls rose their participation in the eagle's diet (adjusted $R^2 = 0.16$, $\beta^2 = 0.44$, $p = 0.03$). The share of categories storks and tortoises (adjusted $R^2 = 0.20$, $\beta^2 = 0.49$, $p = 0.02$) also increased their importance to biomass provision, while lizards and snakes had only a marginal effect through the years (adjusted $R^2 = 0.14$, $\beta^2 = 0.42$, $p = 0.046$). However, other animals also increased their biomass supply (adjusted $R^2 = 0.30$, $\beta^2 = 0.58$, $p = 0.004$).

The strongest negative correlation was found between the categories souslik vs. carnivores ($r_c = -0.61$, $p = 0.002$). Of the other prey categories that had demonstrated a significant trend over the years, hedgehog negatively corelated with water birds ($r_c = -0.50$, $p = 0.02$), carnivores ($r_c = -0.47$, $p = 0.02$), and poultry ($r_c = -0.43$, $p = 0.04$). Increasing stork was related with the depression of poultry ($r_c = -0.49$, $p = 0.02$), carrion ($r_c = -0.48$, $p = 0.02$), and hare ($r_c = -0.45$, $p = 0.03$). The decline of hare also corelated with rise of raptors and owls ($r_c = -0.45$, $p = 0.03$), and that of gulls—with the increasing share of doves ($r_c = -0.47$, $p = 0.02$) and raptors and owls ($r_c = -0.43$, $p = 0.04$). However, another significant negative correlation was found between the categories tortoises and songbirds ($r_c = -0.46$, $p = 0.03$).

The EIE population significantly increased between 1999 and 2021 (adjusted $R^2 = 0.95$, $\beta^2 = 0.97$, $p < 0.001$), starting from eight occupied territories and reaching forty-one in the last year of the study (Figure 2).

### 3.2. Temporal and Spatial Comparison of Eagle Abundance, Profitable Prey Abundance, and Diet Composition in the Studied Periods

Among the three studied periods, the EIE population gradually increased in Sakar Mnt. ($\beta^2 = 0.66$, $p < 0.001$), DHWstr ($\beta^2 = 0.32$, $p < 0.001$), and EYP ($\beta^2 = 0.26$, $p = 0.001$) (Table 3, Figure 4). In contrast, eagle abundance shrank in HM, a process that started in the second period (Tukey's HSD test = 0.045) and was clearly evident in the last one (Tukey's HSD test < 0.001) (Figure 4). In SP, the first pair of EIE occupied the territory in 2007, reaching the maximum number in 2021 (n = 8) (Figure 4). However, in Sakar Mnt., we recorded a significant increase in eagle pairs in the second period (Tukey's HSD test = 0.02), followed by stable population numbers (Tukey's HSD test = 0.90).

**Table 2.** Results of the over-parameterized linear model (GLM) carried out to analyze the trend of the different prey categories (frequency and biomass contribution) of the eastern imperial eagle between 1999 and 2021. We used adjusted $R^2$ value as a correction factor, explanatory parameter estimates ($\beta^2$) with lower (95%) and upper CL (95%), and a probability value (*p*) of the explanatory factors. Significant values are given in bold.

| Prey Categories | Frequency | | | | | Biomass | | | | |
|---|---|---|---|---|---|---|---|---|---|---|
| | Adjusted $R^2$ | $F_{(1.21)}$ | $\beta^2$ | LCL/UCL | *p* | Adjusted $R^2$ | $F_{(1.21)}$ | $\beta^2$ | LCL/UCL | *p* |
| Lizards and snakes | 0.05 | 2.05 | 0.30 | −0.14/0.73 | 0.17 | **0.14** | **4.51** | **0.42** | **0.009/0.83** | **0.046** |
| Tortoises | 0.12 | 4.11 | 0.40 | −0.01/0.82 | 0.055 | **0.20** | **6.62** | **0.49** | **0.09/0.89** | **0.02** |
| Water birds | **0.29** | **10.16** | **−0.57** | **−0.94/−0.20** | **0.004** | **0.14** | **4.46** | **−0.42** | **−0.83/−0.006** | **0.047** |
| Poultry | **0.64** | **40.19** | **−0.81** | **−1.08/−0.54** | **<0.001** | **0.65** | **42.36** | **−0.82** | **−1.08/−0.56** | **<0.001** |
| Phasianids | 0.01 | 0.70 | 0.18 | −0.27/0.63 | 0.41 | 0.01 | 1.16 | 0.23 | −0.21/0.67 | 0.29 |
| Gulls | **0.48** | **21.57** | **−0.71** | **−1.03/−0.39** | **<0.001** | **0.44** | **18.08** | **−0.68** | **−1.01/−0.35** | **<0.001** |
| Doves | **0.33** | **11.91** | **0.60** | **0.24/0.96** | **0.002** | **0.42** | **16.61** | **0.66** | **0.33/1.00** | **<0.001** |
| Songbirds | 0.05 | 0.03 | −0.04 | −0.49/0.41 | 0.86 | 0.05 | 0.02 | 0.03 | −0.42/0.48 | 0.89 |
| Corvids | 0.02 | 0.51 | 0.15 | −0.29/0.60 | 0.48 | 0.04 | 0.18 | 0.09 | −0.36/0.54 | 0.68 |
| Stork | 0.09 | 3.23 | 0.36 | −0.06/0.79 | 0.09 | **0.20** | **6.48** | **0.49** | **0.09/0.88** | **0.02** |
| Raptors and owls | **0.16** | **5.13** | **0.44** | **0.04/0.85** | **0.03** | 0.07 | 2.73 | 0.34 | −0.09/0.77 | 0.11 |
| Hedgehog | **0.34** | **12.41** | **0.61** | **0.25/0.97** | **0.002** | **0.50** | **22.95** | **0.72** | **0.41/1.04** | **<0.001** |
| Hare | **0.67** | **44.79** | **−0.83** | **−1.08/−0.57** | **<0.001** | **0.67** | **46.19** | **−0.83** | **−1.08/−0.58** | **<0.001** |
| Souslik | 0.03 | 0.41 | −0.14 | −0.59/0.31 | 0.53 | 0.04 | 0.20 | −0.10 | −0.55/0.35 | 0.66 |
| Rodentia (excluding souslik) | 0.03 | 0.45 | −0.14 | −0.59/0.30 | 0.51 | 0.05 | 0.01 | −0.005 | −0.46/0.45 | 0.98 |
| Carnivores | 0.01 | 1.16 | −0.23 | −0.67/0.21 | 0.29 | 0.05 | 0.01 | 0.02 | −0.43/0.47 | 0.92 |
| Carrion | **0.35** | **12.70** | **−0.61** | **−0.97/−0.26** | **0.002** | NA | NA | NA | NA | NA |
| Other animals | 0.11 | 3.64 | 0.38 | −0.035/0.80 | 0.07 | **0.30** | **10.47** | **0.58** | **0.21/0.95** | **0.004** |

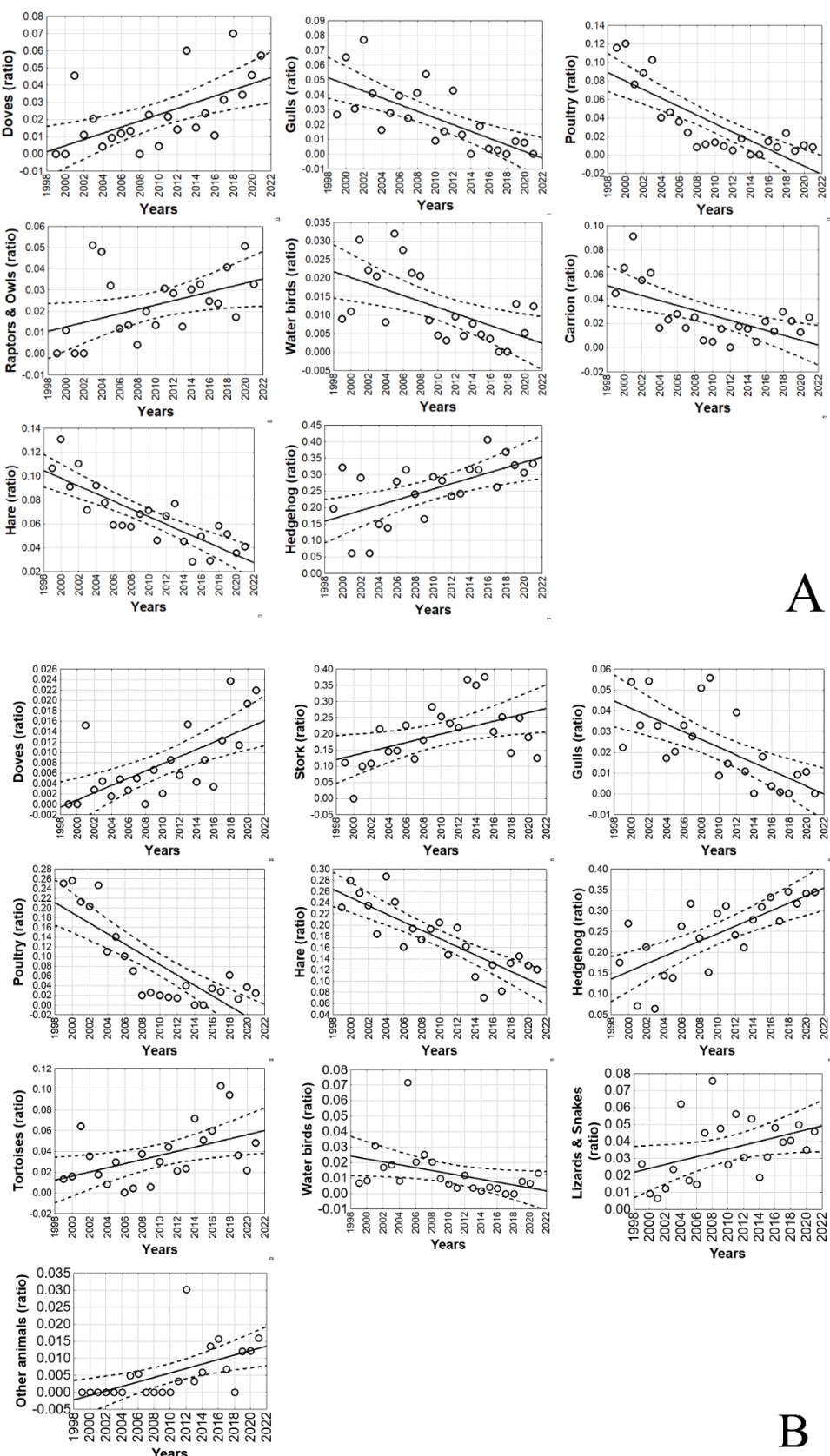

**Figure 3.** Results of the general linear model (GLM) showing a significant trend in frequency (**A**) and biomass (**B**) of different prey categories between 1999 and 2021.

**Table 3.** Results of the general linear mixed model (GLMM) carried out to analyze the trend of souslik abundance and eastern imperial eagle (EIE) abundance in different regions. We used explanatory parameter estimates ($\beta^2$) $\pm$ standard error, with lower (95%) and upper CL (95%) and a probability value (*p*) of the tested categories. Significant values are given in bold.

| Categories | Effect | Region | $\beta^2$ | Std. Err | LCL/UCL | t | *p* |
|---|---|---|---|---|---|---|---|
| **Souslik abundance** | **Random** | **Sakar Mnt.** | **0.46** | **0.20** | **−0.07/0.86** | **2.38** | **0.02** |
| | **Random** | **SP** | **0.38** | **0.18** | **−0.01/0.74** | **2.09** | **0.04** |
| | Random | EYP | 0.30 | 0.18 | −0.06/0.66 | 1.66 | 0.10 |
| | Random | HM | 0.24 | 0.16 | −0.07/0.56 | 1.55 | 0.13 |
| | Random | DHWstr | no tolerance | no tolerance | no tolerance | no tolerance | no tolerance |
| **EIE abundance** | **Random** | **Sakar Mnt.** | **0.66** | **0.08** | **0.51/0.81** | **8.73** | **<0.001** |
| | Random | SP | no tolerance | no tolerance | no tolerance | no tolerance | no tolerance |
| | **Random** | **EYP** | **0.26** | **0.08** | **0.10/0.41** | **3.36** | **0.001** |
| | Random | HM | −0.11 | 0.08 | −0.26/0.04 | −1.44 | 0.15 |
| | **Random** | **DHWstr** | **0.32** | **0.08** | **0.17/0.47** | **4.23** | **<0.001** |

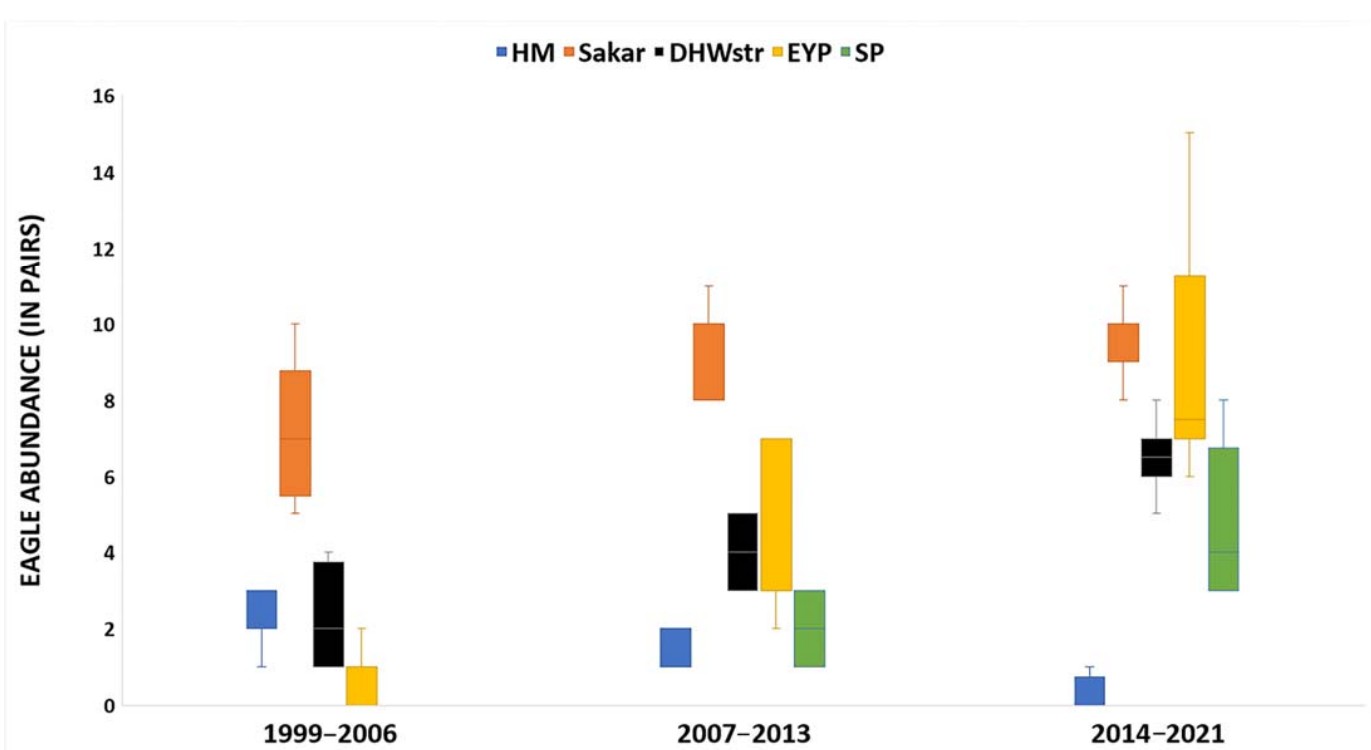

**Figure 4.** Spatial dynamics of the eastern imperial eagle population (Eastern Rhodope Mnt. and Sredna Gora Mnt., HM; Sliven plain, SP; Elhovo-Yambol plain, EYP; Dervent Heights-Western foothills of Strandzha Mnt., DHWstr; Sakar Mnt.).

Abundance of profitable prey sousliks decreased over the study (adjusted $R^2 = 0.25$, $F_2 = 8.97$, $p < 0.001$), a process clearly evident in the third period (Tukey's HSD test = 0.002). A gradual decline was reported in Sakar Mnt. ($\beta^2 = 0.46$, $p = 0.02$) and SP ($\beta^2 = 0.38$, $p = 0.04$), while species' abundance in DHWstr did not show any trend (Table 3, Figure 5). However, souslik abundance in EYP and high mountains (SG and ER) slightly increased in the second period, followed by a severe drop in the last one (Figure 5).

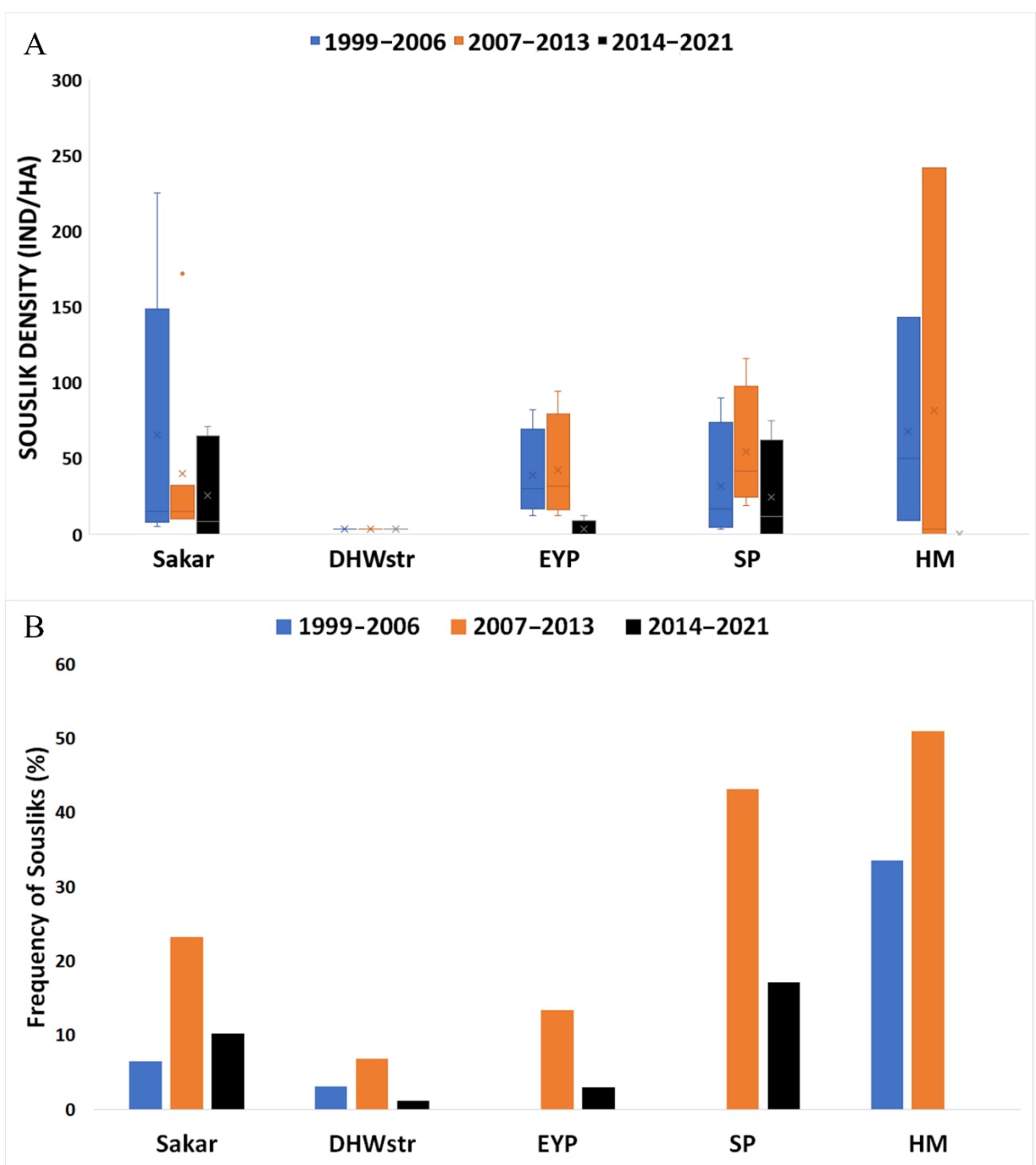

**Figure 5.** Spatial dynamics of souslik abundance (ind/ha) (**A**) and frequency in the eagle's diet (%) (**B**) (Eastern Rhodope Mnt. and Sredna Gora Mnt., HM; Sliven plain, SP; Elhovo-Yambol plain, EYP; Dervent Heights-Western foothills of Strandzha Mnt., DHWstr; Sakar Mnt.).

The frequency of sousliks in the EIE's diet followed more or less the same pattern as that of souslik abundance (Figure 5). In the second period, the presence of this profitable prey marginally increased (Tukey's HSD test = 0.053), followed by a dramatic decline in the last one (Tukey's HSD test = 0.008). In Sakar Mnt., sousliks reached 23.3% of caught prey between 2007 and 2013 and then dropped to 10.21% ($\beta^2 = -0.73$, $p = 0.004$). This prey also shrank significantly its presence in EYP ($\beta^2 = -0.80$, $p = 0.003$). With regard to the decline

of souslik abundance in SP, their frequency was reduced from 43.20% in the second period to 17.16% in the last one. Following the dynamics of the main prey, in high mountains the proportion of sousliks increased in the second period, accounting for 51% of the consumed prey, and then practically disappeared from the eagle's diet in the third one.

Comparing the three studied periods, two prey categories showed obvious significant differences, both with regard to presence and biomass contribution (Table 4). In the second period, lizards and snakes increased their proportion (Tukey's HSD test = 0.05) and biomass participation (Tukey's HSD test = 0.006). This increment was obvious in Sakar Mnt. ($\beta^2 = 1.11$, $p < 0.001$) and DHWstr ($\beta^2 = 0.61$, $p = 0.006$), where this category reached 9.52% and 7.74% of the caught prey, respectively. The other category, water birds, decreased severely in all of the studied regions (Table S1).

One of the most important prey categories, hedgehog, increased its proportion, a process clearly visible in the last period (Tukey's HSD test = 0.03) (Table 4). This phenomenon was observed mostly in Sakar Mnt. in terms of proportion ($\beta^2 = 0.66$, $p = 0.02$) and biomass participation ($\beta^2 = 0.66$, $p = 0.046$). Similarly, in DHWstr, the ratio of hedgehogs in the eagle's diet rose notably ($\beta^2 = 0.70$, $p = 0.01$), starting from 20.54% in the first period and reaching 50.30% in the last one. However, biomass supply by hedgehogs also grew in value from 15.30% to 40.11%, although this change was not statistically significant ($\beta^2 = 0.58$, $p = 0.068$).

Comparing the three studied periods, brown hare (*Lepus europaeus*) also diminished its importance in the EIE's diet (Table 4), a process showing a significant trend in EYP ($\beta^2 = 1.05$, $p = 0.002$), where biomass provided by this prey dropped from 39.08% to 14.90%. Another prey category, rodents, significant declined in the eagle's diet in Sakar Mnt. and DHWstr ($\beta^2 = -0.93$, $p = 0.01$), despite the fact that its meaning to biomass was marginal. However, in HM, the frequency of rodents in the eagle's diet dropped from 49.03% to 18%.

Between the periods, the category stork increased its share (adjusted $R^2 = 0.69$, $F_6 = 5.00$, $p = 0.049$) and biomass participation (adjusted $R^2 = 0.68$, $F_6 = 4.88$, $p = 0.052$) in the EIE's diet (Table 4), but in regional context we observed a significant decline, both in presence ($\beta^2 = 0.99$, $p = 0.01$) and biomass contribution ($\beta^2 = 0.84$, $p = 0.02$), only in DHWstr. In the third period, the eagles breeding in this region reduced the presence of storks in their menu from 17.57% to 10.54%, while biomass provided from storks shrank from 41.9% to 27.2%.

The contribution of songbirds demonstrated different patterns in different regions. They significantly declined in the eagle's diet in Sakar Mnt. ($\beta^2 = 0.78$, $p = 0.004$) and increased in EYP ($\beta^2 = 0.82$, $p = 0.003$) and SP ($\beta^2 = 1.18$, $p < 0.001$). Four other bird prey categories showed regional trends (Table S1). Raptors and owls increased their proportion ($\beta^2 = 1.00$, $p = 0.04$) and biomass supply ($\beta^2 = 1.09$, $p = 0.02$) in EYP, but corvids reduced their share in SP from 9.71% to 5.15% ($\beta^2 = 1.08$, $p = 0.02$), and gulls in DHWstr from 7.95% to 1.39% ($\beta^2 = 0.95$, $p = 0.04$). Phasianids' frequency rose in SP ($\beta^2 = 0.99$, $p = 0.01$) and dropped in EYP ($\beta^2 = 1.15$, $p = 0.005$), although their importance was negligible in general (Table 4). However, in EYP, other vertebrate preys decreased their frequency from 10.58% to 3.81% ($\beta^2 = 0.90$, $p = 0.03$) and their biomass contribution ($\beta^2 = 1.03$, $p < 0.001$).

**Table 4.** Proportion of different eastern imperial eagle prey categories in Bulgaria in the three studied periods.

| Prey Categories | 1999–2006 | | | 2007–2013 | | | 2014–2021 | | | Total (1999–2021) | | |
|---|---|---|---|---|---|---|---|---|---|---|---|---|
| | Number | % N | % Biomass | Number | % N | % Biomass | Number | % N | % Biomass | Number | % N | % Biomass |
| Lizards and snakes | 63 | 4.88 | 2.47 | 154 | 7.82 | 4.83 | 133 | 6.47 | 3.96 | 350 | 6.59 | 3.91 |
| Tortoises | 17 | 1.32 | 2.04 | 28 | 1.42 | 2.20 | 73 | 3.55 | 6.01 | 118 | 2.22 | 3.64 |
| Water birds | 24 | 1.86 | 2.25 | 21 | 1.07 | 1.16 | 11 | 0.54 | 0.47 | 56 | 1.05 | 1.16 |
| Poultry | 77 | 5.97 | 16.53 | 26 | 1.32 | 3.10 | 18 | 0.88 | 2.52 | 121 | 2.28 | 6.22 |
| Phasianids | 24 | 1.86 | 0.87 | 30 | 1.52 | 0.80 | 47 | 2.29 | 1.12 | 101 | 1.90 | 0.94 |
| Gulls | 42 | 3.26 | 2.98 | 57 | 2.89 | 2.99 | 11 | 0.54 | 0.55 | 110 | 2.07 | 2.04 |
| Doves | 12 | 0.93 | 0.29 | 38 | 1.93 | 0.65 | 74 | 3.60 | 1.30 | 124 | 2.33 | 0.81 |
| Songbirds | 50 | 3.88 | 0.30 | 70 | 3.55 | 0.23 | 75 | 3.65 | 0.30 | 195 | 3.67 | 0.27 |
| Corvids | 36 | 2.79 | 1.30 | 64 | 3.25 | 1.50 | 83 | 4.04 | 1.76 | 183 | 3.44 | 1.55 |
| Stork | 59 | 4.57 | 13.88 | 143 | 7.26 | 23.82 | 144 | 7.01 | 23.21 | 346 | 6.51 | 21.11 |
| Raptors and owls | 28 | 2.17 | 1.43 | 35 | 1.78 | 1.14 | 66 | 3.21 | 1.69 | 129 | 2.43 | 1.43 |
| Hedgehog | 237 | 18.37 | 18.46 | 493 | 25.03 | 24.88 | 653 | 31.78 | 31.72 | 1383 | 26.02 | 25.93 |
| Hare | 103 | 7.98 | 22.38 | 123 | 6.24 | 18.03 | 83 | 4.04 | 11.28 | 309 | 5.81 | 16.50 |
| Souslik | 146 | 11.32 | 3.90 | 431 | 21.88 | 7.92 | 141 | 6.86 | 2.44 | 718 | 13.51 | 4.80 |
| Rodents | 262 | 20.31 | 1.07 | 147 | 7.46 | 0.60 | 295 | 14.36 | 0.99 | 704 | 13.25 | 0.87 |
| Carnivores | 55 | 4.26 | 9.58 | 45 | 2.28 | 5.64 | 69 | 3.36 | 9.56 | 169 | 3.18 | 8.14 |
| Carrion | 45 | 3.49 | NA | 24 | 1.22 | NA | 34 | 1.65 | NA | 103 | 1.94 | NA |
| Other animals | 10 | 0.77 | 0.28 | 41 | 2.08 | 0.50 | 45 | 2.19 | 1.13 | 96 | 1.81 | 0.69 |
| **Total** | **1290** | **100.00** | **100.00** | **1970** | **100.00** | **100.00** | **2055** | **100.00** | **100.00** | **5315** | **100.00** | **100.00** |

## 4. Discussion

### 4.1. Long-Term and Large-Scale Changes in the Diet Pattern

Our results clearly demonstrated the prolonged and wide-reaching diet alteration pattern of a generalist top predator. We found that the previously important brown hare and poultry became of less importance, while the northern white-breasted hedgehog (*Erinaceus roumanicus*), white stork (*Ciconia ciconia*), and doves remarkably increased their significance. The ratio of gulls, water birds, and carrion showed a notable decrease, although their roles were marginal. In parallel with the loss of those preys, the categories raptors and owls, lizards and snakes, and tortoises became regularly detected.

In the beginning of the 21st century, poultry and brown hare were found to be the main prey of the EIE in the southeastern part of the country [44]. At the same time, poultry was mentioned as a primary food source for only one pair in ER [30]. We assume that the abandonment of poultry as a food source by eagles was due to a change in bird farming practices and the demographic decline of the population in Bulgaria. Following the disintegration of the communist regime in the country, there was a clear trend of migration of the population to the major cities. Thus few, mostly elderly people, who no longer kept livestock, remained in the small settlements (near which the eagles' nests were located). In addition, a strict order introduced after 2006, related to cases of avian influenza and banning of free-range poultry farming, severely limited the possibilities for eagles to catch such prey.

The decline of brown hare in the EIE's diet corresponded to the reported decrease of the species' population in Bulgaria in the past decades, especially for EYP [45]. The significant transformation of grasslands [41] expressed in the removal of shrub vegetation through shredders and bulldozers shrank the optimal habitats for hares. However, the population crash of brown hare due to different epizootic diseases was also an important factor affecting the species' abundance and availability [46,47].

In contrast to the Pannonian population of the EIE [24], water birds and carrion remarkably reduced their ratio in the EIE's diet. After the country's accession to the EU (2006) and the introduction of strict sanitary regulations concerning carcass disposal [48], carrion became less frequented in the eagle's diet. The reduction of water birds was probably associated with a decrease in their abundance. It was recorded that some colonies of herons (Ardeidae) distributed along the lower reaches of the Tundzha river disappeared or reduced in number. On the other hand, wildfowl (Anatidae) that were more frequently predated in the winter period [22] decreased their abundance in the study area (author's data).

The reduction of gulls in the EIE's food spectrum was probably related to the reduction of their abundance, which could be a result of the elimination of unregulated landfills, concentrating large flocks of birds. However, yellow-legged gulls (*Larus michahelis*) are still one of the main food sources for eagles in the neighboring population of European Turkey [23].

The substitution of brown hare, water birds, and Poultry in some regions by northern white-breasted hedgehog and white stork could hardly be associated with a sharp increase in the abundance of these substitute prey species. The substitute prey probably existed in the territories of the eagles with similar abundance, but the eagles met their nutritional needs predating hares and easy catches, such as poultry species, which, being in significant quantities, represented a more nutritious source of biomass. Therefore, if these species decreased, eagles had to switch to another, less nutritious yet plentiful food source, such as hedgehogs, or more difficult to capture but with more biomass, such as white storks. However, this issue needs further clarification. Nevertheless, the described drastic and large-scale transformation in grassland habitats [41] and the direct mass extermination of hedgehogs by fast-moving shredders may soon lead to the depletion of this favorable food source. However, for how long hedgehogs will remain suitable prey for the EIE is highly questionable since habitat suitability is expected to become less favorable when habitat transformation affects large areas.

The shift from gulls to doves, the increased proportion of tortoises, and the intraguild predation were probably related to the eagles' adaptation to different food sources. However, individuals ranked these subsequent prey species differently, which was in line with the competitive refuge model according to the optimal foraging theory (OFT) [12,49,50]. In any case, these circumstances need further research.

*4.2. Profitable Prey Abundance Changes and Adaptive Response of EIE*

We found that the abundance of profitable prey for the EIE, such as sousliks, depleted trough the studied periods. This was clearly evident in the last period for most of the studied regions, except DHWstr, where souslik availability and abundance were very scarce and where its presence in the eagles' diet was less than 4% [22]. However, the decrease of souslik could be associated with the vast habitat alteration reported for the EIE distribution range in Bulgaria [41]. It is crucial to understand the particular diet response of eagles to the habitat changes in each occupied territory as well as whether this reaction depends on the size or any other characteristics of the favorable habitat. Anyway, this issue should be the focus of future research.

Our expectation that the presence of sousliks in the eagle's diet would follow the dynamics' pattern of this prey was confirmed. Despite the severe decline of the profitable prey, such as sousliks, the EIE population benefited in most of the studied regions, evidence for successful adaptation of this top predator. Our study confirmed previous findings, namely that the EIE was able to alter its diet and utilize the most available and/or abundant prey sources [22,24]. The significant shifting towards hedgehogs, white storks, pigeons, tortoises, and birds of prey was a good example of the successful adaptation of the EIE to a novel and accessible food source. Similar adaptation is known for another large top predator, the golden eagle (*Aquila chrysaetos*), which substitutes hedgehogs for its favorite prey, tortoises [16]. However, eagles could only shift and survive in those territories where their main prey decreased if alternative species were available and sufficiently abundant. For example, in parallel with the decrease of souslik populations, eagles' abundance also gradually declined in mountain regions (SG, ER). In fact, the last known mountain EIE's territory has been unoccupied since 2016. We speculate that the availability and abundance of substitute prey, such as hedgehogs, storks, and pigeons, was not enough to secure and sustain the birds, hence they abandoned these territories. Depression in the EIE population due to souslik degradation was reported in different regions of Russia [51,52]. However, this issue needs further confirmation.

Conversely, the decrease of sousliks, brown hare, and poultry in the rest of the study area forced eagles to prey more intensively on hedgehogs or forage for substitute prey, such as white stork, different reptiles, diurnal and nocturnal raptors, or songbirds. As a consequence, the EIE population expanded or remained stable in this part of the distribution area.

The trophic strategy used by eagles towards opportunistic foraging is an ecological advantage that allows the species to adapt to different habitats. According to the alternative prey hypothesis (APH), [53], a generalist predator such as the EIE may synchronize its diet with the fluctuations of main and alternative prey groups.

*4.3. Effects of Diet Alteration and Conservation Suggestions*

The importance of brown hare and the presence of small game species (phasants, partridge) in the prey of the EIE stirred a significant negative attitude among hunters towards eagles [24]. This effect intensified the "human–predator" conflict and led to human-related mortality due to persecution. Illegal shooting accounted for 12.5% of EIE mortality in Bulgaria [54], and there is evidence that this threat is increasing. It is crucial to communicate actively with and raise the conservation awareness of hunters. Improved communication between conservationists and hunters is known to be effective both in reducing violations and recognizing the mutual interest in lobbying for environmentally friendly practices in agricultural land use [24].

The increasing frequency of feral pigeon (*Columba livia var. domestica*) in the EIE's diet can also raise conflicts with pigeon fanciers, which in turn could result in persecution incidents. An immature eagle tagged with a satellite transmitter was poisoned through a bait set by pigeon fanciers in the second most important temporary settlement site of the species in the country [55]. In this area, poison baits set by pigeon fanciers cause mortality of different raptors such as long-legged buzzard (*Buteo rufinus*), peregrine falcon (*Falco peregrinus*), and saker falcon (*Falco cherrug*) [55,56].

The eagles' predation on poultry species, particularly intensive in the first study period, could also raise conflicts with poultry keepers, which would result in persecution incidents.

Feeding on carrion poses a potential threat of poisoning due to illegal baits used to control predators. Poisoning was identified as the most important mortality factor affecting the breeding population of the EIE in Bulgaria [54].

## 5. Conclusions

We found long-term and large-scale diet alterations of EIE. While brown hare, poultry, gulls, water birds, and carrion decreased over the years, northern white-breasted hedgehog and doves increased both in frequency and biomass provision. Raptors and owls raised their participation, but white stork and different reptiles supplied more biomass.

The abundance of European souslik decreased through the studied periods, which accounted for the lower proportion of this prey species in the eagle's diet. Nevertheless, the EIE population successfully adapted and significantly increased in most of the distribution area. Our idea that eagles could survive and expand in territories where their profitable prey decreased only if alternative species were available and abundant, was indirectly confirmed. The observed adaptive plasticity through alterations of the EIE's diet in response to temporal and spatial prey changes greatly facilitates conservation efforts, as it seems that although the species feeds on the most abundant prey, it does not depend solely on the state of any particular source of food. Therefore, conservation efforts should focus on the preservation of its main foraging habitats and the restoration of damaged ones so as to maintain a good condition of both its main food source in the area and the subsequent prey. Predator–prey interactions and conservationists–stakeholders conflict management are crucial for the effective preservation of this endangered top predator.

**Supplementary Materials:** The following supporting information can be downloaded at: https://www.mdpi.com/article/10.3390/d14111000/s1, Table S1: Results of the General Liner Mixed Model (GLMM) carried out to analyze the trend of the different prey categories (frequency and biomass) in different regions.

**Author Contributions:** Conceptualization, D.D. (Dimitar Demerdzhiev) and D.D. (Dobromir Dobrev); methodology, D.D. (Dimitar Demerdzhiev) and D.D. (Dobromir Dobrev); software, D.D. (Dimitar Demerdzhiev) and D.D. (Dobromir Dobrev); validation, D.D. (Dimitar Demerdzhiev), Z.B., D.D. (Dobromir Dobrev), N.N. and T.P.; formal analysis, D.D. (Dimitar Demerdzhiev); investigation, D.D. (Dimitar Demerdzhiev), Z.B., D.D. (Dobromir Dobrev), N.N. and T.P.; resources, D.D. (Dimitar Demerdzhiev); data curation, D.D. (Dimitar Demerdzhiev) Z.B., D.D. (Dobromir Dobrev), N.N. and T.P.; writing—original draft preparation, D.D. (Dimitar Demerdzhiev); writing—review and editing, Z.B., D.D. (Dobromir Dobrev), N.N. and T.P.; visualization, D.D. (Dimitar Demerdzhiev) and D.D. (Dobromir Dobrev); supervision, D.D. (Dimitar Demerdzhiev), D.D. (Dobromir Dobrev) and Z.B.; project administration, D.D. (Dobromir Dobrev); funding acquisition, D.D. (Dobromir Dobrev). All authors have read and agreed to the published version of the manuscript.

**Funding:** This work was partially funded by the LIFE Program of the European Union under the project 'Restoration and sustainable management of Imperial Eagle's foraging habitats in key Natura 2000 sites in Bulgaria' LIFE14 NAT/BG/001119.

**Institutional Review Board Statement:** Not applicable.

**Data Availability Statement:** Not applicable.

**Acknowledgments:** We would like to thank Georgi Popgeorgiev, Vladislav Vergilov, Nikolay Tzankov, and Andrey Stoyanov for their assistance with the identification of the remains of amphibians and reptiles; Vassil Popov and Nikolay Spasov for the identification of some mammalian remains; Mladen Jivkov and Tihomir Stefanov for the identification of fish prey. Our special thanks go to Nikolay Terziev for nest climbing and collection of a part of the food remains. We are grateful (alphabetically) to Atanas Delchev, Atanas Demerdzhiev, Aleksandar Georgiev, Dimitar Plachiyski, Georgi Georgiev, Georgi Gerdzhikov, Hristo Hristov, Iliya Iliev, Ivaylo Angelov, Kiril Metodiev, Krasimir Andonov, Krasimira Demerdzhieva, Milan Bakalov, Stefan Avramov, Stoycho Stoychev, Svetoslav Spasov, Vanyo Angelov, Vera Dyulgerska, Vladimir Dobrev, Vladimir Trifonov, and Volen Arkumarev, who took part in the field work. Without their assistance this survey would not have been possible. We are grateful to two anonymous reviewers who improved the draft of the manuscript.

**Conflicts of Interest:** The authors declare no conflict of interest.

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
