# Peer review of "Does Temporal and Spatial Diet Alteration Lead to Successful Adaptation of the Eastern Imperial Eagle, a Top Predator?"

_diversity, doi:10.3390/d14111000_

Round 1
Reviewer 1 Report
This is an interesting paper on the dietary response of the top predator Eastern Imperial Eagle on the changes in the availability of its main prey. There are several strong points of the work, including the unique data set that covers two decades of documenting prey remains and the application of General Linear Models. The reasoning and the methodology of this study are solid.
In my opinion, the manuscript is already good and should be published after considering the points -mentioned below:
Page 15 and 38 – explain what are “population trajectories
Page 24, 439 – the statement that “The trophic strategy… guarantees its future” sounds very optimistic, rather without a scientific base
Page 59, 68, etc – needs correction of font size
Page 85 – remove “extreme”
Page 146, 150 – replace “vs.” with “and”
Page 222 “main prey dynamics” suggests the presence of data on prey populations, but the chapter describes changes in prey contribution to the diet. Please change
Page 224 (and the whole subchapter) – “While categories Hare, Poultry and Gulls showed largest decrease” and similar sentences below should be changed to „The share/contribution of categories Hare, Poultry and Gulls decreased in the diet…” . The categories did not decreased, reduced, increased (eg Increasing Stork … page 238), etc, these all were the changes in EIE diet!
Page 246 – Please check the order of citation of figures in the paper
Page 274, 294 – Please build a proper figure legend
Page 298-300 – Lizards and Snakes form one prey category, not two
Page 313 – Please use scientific names for the other prey species, not only the Hare
Page 326 “Song birds demonstrated different patterns in different regions”. The authors probably mean the different contributions of song birds to the EIE diet! Please change
Page 337 The Table is not about „Distribution of different …, Please change.
Author Response
Dear Reviewer,
thank you for your comments and suggestions in the manuscript. We carefully checked all of them and accepted all your suggestions.
Please find our replies attached bellow.

Reviewer 2 Report
General Comments
The paper entitled "Does temporal and spatial diet alteration lead to successful adaptation of a top predator" (MS number: diversity-1993926) is quite interesting for the effective preservation of the endangered Eastern Imperial Eagle (Aquila heliaca) in Bulgaria. As far as I can understand, it is the first paper that actually evaluates large-scale predator-prey interactions of the latter top predator by using a long term (23 years) data set.
In general, ms is well structured and refer to other relevant studies. Data set as well as methodology used are adequate to sustain discussion and conclusions. I only have minor edits as indicated below.
Materials and Methods
Lines 96-97: Provide reference(s)
Lines 157-161: I feel that you have to give some more clarifications for the sampling method (e.g. what is actually the effect of cycles in rodent abundance between years?)
Discussion
Line 348: Use “21st century” instead of “21th century”
Author Response
Dear Reviewer,
thank you for your comments and suggestions. We changed and added what you proposed.
You can find our responses attached bellow.
